# Dynamical Mechanism Analysis of Three Neuroregulatory Strategies on the Modulation of Seizures

**DOI:** 10.3390/ijms232113652

**Published:** 2022-11-07

**Authors:** Honghui Zhang, Zhuan Shen, Yuzhi Zhao, Lin Du, Zichen Deng

**Affiliations:** 1School of Mathematics and Statistics, Northwestern Polytechnical University, Xi’an 710072, China; 2MIIT Key Laboratory of Dynamics and Control of Complex Systems, Xi’an 710072, China; 3School of Aeronautics, Northwestern Polytechnical University, Xi’an 710072, China

**Keywords:** Wilson–Cowan model, optogenetic stimulation, deep brain stimulation, electromagnetic induction, epilepsy

## Abstract

This paper attempts to explore and compare the regulatory mechanisms of optogenetic stimulation (OS), deep brain stimulation (DBS) and electromagnetic induction on epilepsy. Based on the Wilson–Cowan model, we first demonstrate that the external input received by excitatory and inhibitory neural populations can induce rich dynamic bifurcation behaviors such as Hopf bifurcation, and make the system exhibit epileptic and normal states. Then, both OS and DBS are shown to be effective in controlling the epileptic state to a normal low-level state, and the stimulus parameters have a broad effective range. However, electromagnetic induction cannot directly control epilepsy to this desired state, even if it can significantly reduce the oscillation frequency of neural populations. One main difference worth noting is that the high spatiotemporal specificity of OS allows it to target inhibitory neuronal populations, whereas DBS and electromagnetic induction can only stimulate excitatory as well as inhibitory neuronal populations together. Next, the propagation behavior of epilepsy is explored under a typical three-node feedback loop structure. An increase in coupling strength accelerates and exacerbates epileptic activity in other brain regions. Finally, OS and DBS applied to the epileptic focus play similar positive roles in controlling the behavior of the area of seizure propagation, while electromagnetic induction still only achieves unsatisfactory effects. It is hoped that these dynamical results can provide insights into the treatment of epilepsy as well as other neurological disorders.

## 1. Introduction

Epilepsy is a central nervous system disorder in which brain activity becomes abnormal, causing periods of unusual behavior, sensations and sometimes loss of awareness. The current common treatments for it include anti-seizure medication, epilepsy surgery and neuroregulatory therapy. The applicabilities and disadvantages of these methods make the treatment of epilepsy a challenging scientific research topic.

In terms of neuroregulatory strategies, electrical and magnetic technologies have been the research focus over the past century. As a perfect representative of electrical stimulation, deep brain stimulation (DBS) can control the stimulation target by changing the frequency, width and amplitude of the pulse signal to alleviate neurological diseases and movement disorders. Since it was proposed, DBS has achieved considerable results in animal experiments [1,2,3], clinical practice [4,5,6] and theoretical research [7,8,9]. In order to reduce the side effects and improve the applicability of stimulation strategies, multi-target combination and intermittent forms of DBS have also been proposed [7,10,11]. Aside from concerns about power overhead and electrode degradation, DBS is definitely an invasive method.

One typical non-invasive brain function modulation technology is transcranial magnetic stimulation (TMS). By applying short pulses of magnetic fields to the cortex through current-carrying coils, TMS can induce electrical currents in the cerebral cortex, affecting activity in a large area of the brain (several square centimeters) [12]. Studies have shown that TMS may become a new epilepsy treatment [13,14]. However, the effect mechanism of TMS is still unclear, and the treatment is also controversial. In order to study the effects of electromagnetic fields, scholars have explored the changes of membrane polarization voltage induced by electromagnetic fields based on neuron models, summarized the response mechanisms of neurons, and explored the effects of electromagnetic disturbances on the firing activity of individual neurons and neuron networks [15,16,17,18]. It was also observed experimentally that fluctuations in neuronal membrane potential can induce time-varying magnetic fields [19]. However, how electromagnetic induction affects the development of epilepsy has not yet been fully explored from the perspective of dynamic.

Studies have shown that inhibitory neurons play an important role in epileptic seizures [20,21,22], but it is difficult for DBS and TMS to target a specific neurons in the cortex (especially the inhibitory neurons) without affecting peripheral neurons. In recent years, optogenetics technology has gradually developed to effectively solve such problems. Optogenetic stimulation (OS) selects a virus as a carrier to import genes encoding light-sensitive proteins into specific target cells, which can precisely excite or inhibit neurons after being irradiated with different wavelengths of exogenous light [23]. Whether controlling inhibitory neuronal populations can control seizures remains unknown. Although there have been many meaningful experimental and theoretical studies giving a positive answer [24,25,26,27], this is still far from clinical practice.

Inspired by these, we are interested in further exploring the role of OS, DBS and electromagnetic induction in epilepsy, including its generation, propagation and termination. More importantly, we try to compare the three regulatory mechanisms from the perspective of theoretical models and dynamics. It should be noted that the current dynamics models for studying neuronal activity are mainly from the micro [28,29,30,31] and macro levels [32,33,34,35,36,37]. For epilepsy research, macroscopic models are used more frequently. The Wilson–Cowan model [32], as the first mathematical model to emphasize the significance of interactions between excitatory and inhibitory neural populations, has inspired the development of epilepsy models [33,34,35,36,37]. Therefore, the Wilson–Cowan model has also become the first choice for the basic model of the regulation research in this paper.

The rest of the content is structured as follows: the descriptions of the Wilson–Cowan model, OS, DBS and electromagnetic induction are given in Section 4. The main results, including the modulation of epileptiform activity by the model external inputs, three strategies, and three strategies in the three-node feedback loop structure, are presented in Section 2. Discussions are drawn in Section 3.

## 2. Results

### 2.1. Epileptiform Activities Regulated by the Model External Inputs

We first investigated the dynamical behaviors induced by external inputs $P_{E}$ and $P_{I}$ in order to better carry out the follow-up studies. The results show that $P_{E}$ and $P_{I}$ can induce rich transition phenomena. As shown in Figure 1a, when fixing $P_{I}=0.25$ and changing $P_{E}$, the system will experience four bifurcations, namely: fold of fixed point bifurcations (LPF${}_{1}$ (1.037, 0.1141), LPF${}_{2}$ (1.106, 0.0563)) and Hopf bifurcations (HB${}_{1}$ (1.064, 0.135), HB${}_{2}$ (1.896, 0.2233)). When the value of $P_{E}$ belonged to region I, the dynamical system was at the stable fixed point state, that is, the firing rates of E and I were all low. Since the resting state (E(t)=0 and I(t)=0) was taken to be a state of low-level background activity, the ubiquitous situation in neural tissue [32], and the state in region I were desired. Its time series and phase diagram are shown in Figure 1d ($P_{E}=0.75$, $P_{I}=0.25$) and Figure 1f, respectively. As $P_{E}$ increases (i.e., $P_{E}\in \left[1.037,1.064\right]$, region II), LPF${}_{1}$ causes a bistable region composed of two stable fixed points. With the further increase of $P_{E}$ (i.e., $P_{E}\in \left[1.064,1.106\right]$, region III), a stable limit cycle comes into being because of the occurrence of HB${}_{1}$, which makes the system turn into a bistable region consisting of one stable fixed point and limit cycle. Then, when the system goes through the second fold of fixed point bifurcation (LPF${}_{2}$ (1.106, 0.0563)), the stable focus becomes unstable and the system only remains in the stable limit cycle (i.e., $P_{E}\in \left[1.106,1.896\right]$, region IV). So the system exhibits an oscillating state; its time-series and phase diagram are also shown in Figure 1d ($P_{E}=1.25$, $P_{I}=0.25$) and Figure 1e, respectively. Afterward, the second Hopf bifurcation (HB${}_{2}$ (1.896, 0.2233)) causes the stable limit cycle to lose stability, and the system is in a monostable state with only one stable fixed point (i.e., $P_{E}\in \left[1.896,2\right]$). It is important to note that the equilibrium point state in region V is not ideal because it leaves the neural tissue in a high level of continuous oscillation. However, the medium-level stable equilibrium state (i.e., E∈[0.1141,0.135]) in region II is better than the oscillatory state to a certain extent.

When fixing $P_{E}=1.1$, the bifurcation analysis induced by $P_{I}$ is shown in Figure 1b. It is clear that the dynamical system is initially at the monorhythmic state ($P_{I}\in \left[0,0.1982\right]$, region IV), so the system presents an oscillatory state in the time domain. By increasing $P_{I}$ ($P_{I}\in \left[0.1982,0.2852\right]$, region III), the fold of the fixed point bifurcation (LPF${}_{1}$ (0.1982, 0.05521)) appears, which leads to the coexistence of stable limit cycles and low-level stable equilibrium points. When the system goes through the Hopf bifurcation HB(0.2852, 0.1376), the stable limit cycles become unstable and the medium-level stable equilibrium state appears. So the low-level stable equilibrium state and the medium-level stable equilibrium state are coexisting in this parameter range ($P_{I}\in \left[0.2852,0.3801\right]$, region II). With the further increase of $P_{I}$ ($P_{I}\in \left[0.3801,2\right]$, region II), the second fold of the fixed point bifurcation (LPF${}_{2}$ (0.3801, 0.1004)) is experienced. The medium-level equilibrium state disappears, and the system only presents a monostable low-level equilibrium state. In other words, when $P_{I}$ takes a large value ($P_{I}\in \left[0.3801,2\right]$, region II), the system is always in a normal physiological state.

Overall, we focus on the combined modulation of $P_{E}$ and $P_{I}$. It can be clearly seen from Figure 1c that the system always presents five states when $P_{E}\in \left[0,2\right]$ and $P_{I}\in \left[0,0.5\right]$. The meanings represented by states I, II, III, IV and V are the same as in Figure 1a,b. The dominant states on the right-hand side of the plane are IV and V, while the low-level equilibrium state is located on the left (I). This indicates that, when the value of $P_{E}$ is larger, the system presents the oscillating state as well as the high-level equilibrium state. In other words, reducing the value of $P_{E}$ is beneficial to the system to recover from the pathological state to the normal state. In addition, the existence of bistable region III implies that the application of appropriate regulatory strategies can prompt the system to transition between the stable limit cycle and the low-level stable equilibrium state. The theoretical dynamic analysis in this section lays the foundation for our subsequent research on regulatory strategies.

### 2.2. Epileptiform Activities Regulated by the Three Strategies

Based on the classical Wilson–Cowan model, in this section, we focus on the effects of OS, DBS and electromagnetic induction on epilepsy. The first is the regulation effect of OS. Let the system be in the pathological state initially ($P_{E}=1.25$, $P_{I}=1.05$ ); there are four typical regulation situations as shown in Figure 2a–d. The bottom black lines represent the illumination strategy in the form of pulses, which is added to the system at t = 200 ms. Specifically, fixing ws = 5 ms, f = 40 Hz, OS targeting I has a limited effect on the system (Figure 2a, I${}_{rr}$ = 0.010 mW/mm${}^{2}$). As I${}_{rr}$ increases to 0.020 mW/mm${}^{2}$, E and I both show a low-amplitude oscillation (peak-trough difference is less than 0.05) with a frequency of 40 Hz, which is due to the accurate response of neurons to OS. Notably, it is the ideal state we are pursuing. Interestingly, when I${}_{rr}$ gradually increases to 0.080 mW/mm${}^{2}$, E continues to oscillate at a low amplitude, while I eventually exhibits large-amplitude oscillation. In other words, the high-intensity light stimulation overexcites I making it show the pathological state again. A constant light stimulus is also considered in Figure 2d (f = 40 Hz, ws = 25 ms, I${}_{rr}$ = 0.080 mW/mm${}^{2}$). It can be seen that E transitions to a low-level state, while I transitions to a high-level stable equilibrium state (pathological state). Although E in Figure 2c,d shows a normal state after OS, I appears to have different degrees of pathology. OS is ineffective in both cases. Combining the observed quantitative indicators (peak, trough, peak-trough of the stable time series of E and I), Figure 2e presents the regulation effect of I${}_{rr}$ in detail. Clearly, both E and I are in large-amplitude oscillations, and the system behaves in the epileptic state (I${}_{rr}$ < 0.012). With I${}_{rr}$ increasing to around 0.015 (optimal stimulus intensity), a clear turning point appears, indicating that both E and I are low-amplitude oscillations. For larger I${}_{rr}$, E still maintains a low-amplitude oscillation, while the oscillation amplitude of I keeps increasing.

A more comprehensive dynamic response under the influence of dual stimulation parameters is shown in Figure 3. Here, f = 40 Hz (Figure 3a,d), I${}_{rr}$ = 0.015 mW/mm${}^{2}$ (Figure 3b,e) and ws = 2 ms (Figure 3c,f) are fixed. The area enclosed by the white dashed line in Figure 3 is the effective parameter area (peak, trough, peak-trough of E and I are all smaller than 0.05). Taking Figure 3a,d as the examples, when I${}_{rr}$ is small, E and I are in the high amplitude pathological oscillation regardless of how ws is changed. As I${}_{rr}$ increases, a dark blue region appears in the plane, i.e., the peak–trough of both E and I decreases, indicating that light stimulation within this parameter range could effectively modulate the pathological mode. As I${}_{rr}$ continues to increase across the white dashed range, the peak–trough of E remains at a low level, while that of I gradually increases, so the system is again in the pathological state. We need to make it clear that the blue area in the upper right corner of the figure is discarded because it is at a high-level stable equilibrium state (i.e., the system is still characterized by pathology). In conclusion, a moderate light stimulation parameter setting is superior.

The regulation effect of DBS is similar to that of OS. It is impossible to stimulate the inhibitory neuronal population I without affecting the excitatory neuronal population E in actual operation, so DBS is applied to both E and I. Additionally, the black lines represent DBS in pulsed form, and we applied it at t = 200 ms as shown in Figure 4. The results show that the system will present three states: E and I both maintain high-amplitude, ill-conditioned oscillation (Figure 4a, δ = 2 ms, f${}_{DBS}$ = 60 Hz, I${}_{DBS}$ = 2 mA). E is a low amplitude oscillation and I is an almost normal low-level state (Figure 4b, δ = 2.5 ms, f${}_{DBS}$ = 150 Hz, I${}_{DBS}$ = 1.8 mA); both E and I are almost at a normal low-level state (Figure 4c, δ = 2 ms, f${}_{DBS}$ = 190 Hz, I${}_{DBS}$ = 2 mA ). Among them, the control effect in Figure 4a indicates that the stimulus is ineffective, while that in Figure 4b,c shows that the stimulus is effective. The latter two stimulation outcomes are the desired situations when applying the DBS. In particular, unlike light stimulation, there is no high-level stable equilibrium state under the DBS situation.

Similarly, we also present the regulatory effect of DBS’ parameters on neuronal activities in Figure 5. The dark blue area represents the stimulus effective area. Take the first column as an example (Figure 5a,d), if δ≤ 2 ms, no matter how f${}_{DBS}$ changes, E and I are always in the ill-conditioned oscillation. As 2 ms ≤δ≤ 3.5 ms, the nf${}_{DBS}$ required to suppress pathologic oscillation decreased with the increase of δ. When δ > 3.5 ms, the ill-conditioned oscillation can always be effectively controlled regardless of f${}_{DBS}$, that is, the system is in low-amplitude oscillation or a low saturated state. It can be seen that δ seems to play a more important role from the perspective of regulatory effect.

The situation under the influence of electromagnetic inductance is shown in Figure 6 and Figure 7. Similar to the case of OS and DBS, the initial state of the system is set to the oscillating state by fixing $P_{E}$ and $P_{I}$. The electromagnetic induction model is coupled with the Wilson–Cowan model at t = 200 ms as indicated by the $I_{mf}$ and arrow in the Figure 6. Clearly, if the magnetic feedback effect is very weak ($K_{0}=0.01$, Figure 6a), the state of the system is hardly affected. When $K_{0}$ is taken to a larger value ($K_{0}=0.05$, Figure 6b), the existence of electromagnetic induction will reduce the amplitude of the original oscillation. When $K_{0}$ is larger ($K_{0}=0.1$, Figure 6c), the oscillatory state transitions to a higher level of background state. However, in any case, the firing rate represented by the equilibrium state is generally smaller than that of the oscillating state (Figure 6c). The oscillating state can also be regulated to the medium-level stable equilibrium state ($K_{0}=0.5$, Figure 6d).

Figure 7 explains this phenomenon induced by electromagnetic induction from the view of dynamics. Compared with Figure 1a, the results in Figure 7a show that weak electromagnetic induction does not induce significant changes in the dynamic transition behavior of the system. It is only manifested as a slight left shift of LPF${}_{1}$ (from (1.037, 0.1141) to (1.042, 0.1116)), LPF${}_{2}$ (from (1.106, 0.0563) to (1.095, 0.0584)) and HB${}_{2}$ (from (1.896, 0.2233) to (1.867, 0.2211)), and a right shift of HB${}_{1}$ (from (1.064, 0.135) to (1.092, 0.2211)). So, the region of the oscillating state is slightly reduced. When the effect of electromagnetic induction is stronger, in addition to the movement of the bifurcation points, the most obvious is that the range and size of the stable limit cycle are reduced. It suggests that proper electromagnetic induction can reduce the amplitude of the oscillating state (Figure 7b). Surprisingly, when the electromagnetic induction feedback continues to increase, all the bifurcation points disappear and the system only behaves as a monostable equilibrium point state (Figure 7b). At the same time, we change the value of $P_{I}$ from 0.25 to 0.01, and still observe the dynamic bifurcation transition phenomenon of the system under different electromagnetic induction strengths (Figure 7d–f). The phenomenon is similar to that shown in Figure 7a–c, and we will not repeat it. Furthermore, we analyzed changes in the two-parameter plane including bifurcation points and stable regions. Obviously, the increase of $K_{0}$ undoubtedly reduces the area of the oscillation region (IV). However, it is worth noting that electromagnetic induction is different from OS and DBS. Electromagnetic induction can hardly regulate the ill-conditioned oscillatory state to a normal low-level state.

### 2.3. Propagation of Epileptiform Activities in the Coupled Network Structure

Network motifs in the brain play an important role in the study of neurological diseases such as epilepsy [38,39,40]. Among them, the most frequent and the most widely studied is the three-node network motif structure [38]. Based on the three-node feedback loop structure, one of the typical three-node network motifs, we further explore the regulatory mechanisms of the three regulatory strategies on the propagation of epileptiform activities in the next two sections. In this section, we present the epileptiform propagation behavior under the three-node coupled network model.

The three-node coupled Wlison-Cowan network model is as below:(1)$$\begin{array}{ccc} \tau_{E}\frac{dE_{1}}{dt} & =-E_{1}+\left(k_{E}-r_{E}E_{1}\right)S_{E}\left(C_{1}E_{1}-C_{2}I_{1}+P_{E_{1}}+KE_{3}\right), & \end{array}$$(2)$$\begin{array}{ccc} \tau_{I}\frac{dI_{1}}{dt} & =-I_{1}+\left(k_{I}-r_{I}I_{1}\right)S_{I}\left(C_{3}E_{1}-C_{4}I_{1}+P_{I_{1}}\right), & \end{array}$$(3)$$\begin{array}{ccc} \tau_{E}\frac{dE_{2}}{dt} & =-E_{2}+\left(k_{E}-r_{E}E_{2}\right)S_{E}\left(C_{1}E_{2}-C_{2}I_{2}+P_{E_{2}}+KE_{1}\right), & \end{array}$$(4)$$\begin{array}{ccc} \tau_{I}\frac{dI_{2}}{dt} & =-I_{2}+\left(k_{I}-r_{I}I_{2}\right)S_{I}\left(C_{3}E_{2}-C_{4}I_{2}+P_{I_{2}}\right), & \end{array}$$(5)$$\begin{array}{ccc} \tau_{E}\frac{dE_{3}}{dt} & =-E_{3}+\left(k_{E}-r_{E}E_{3}\right)S_{E}\left(C_{1}E_{3}-C_{2}I_{3}+P_{E_{3}}+KE_{2}\right), & \end{array}$$(6)$$\begin{array}{cc} \tau_{I}\frac{dI_{3}}{dt} & =-I_{3}+\left(k_{I}-r_{I}I_{3}\right)S_{I}\left(C_{3}E_{3}-C_{4}I_{3}+P_{I_{2}}\right). \end{array}$$

Here, *K* is the coupling connection strength. We set the coupling strength between the three cortical columns to be the same for simplicity; in fact, they can be different. The meanings and values of other parameters are the same as those in Section 4.1.

Firstly, we set $P_{E_{1}}=1.25$, $P_{E_{2}}=P_{E_{3}}=1.05$, that is, column1 is in an oscillating state (epileptiform state), while column2 and column3 are in the normal background state (ideal state). And the column1, column2 and column3 are not coupled to each other (K=0, Figure 8a). With the increase of *K*, column2 and column3 gradually transitioned from the background state to the epileptic state. As shown in Figure 8b, when K=0.22, column2 becomes epileptic at t = 628 ms, and column3 remains background state. When *K* is further increased (K=0.31, Figure 8c), both column2 and column3 become epileptic, but the onset of oscillations in column3 (t = 883 ms) is significantly later than that in column2 (t = 202 ms). When K=0.52 (Figure 8d), this phenomenon is more obvious, but the time interval between the onset of column2 and column3 is shortened (for column2: t = 103 ms, for column3: t = 169 ms). It is worth noting that the frequency of the oscillations of column2 and column3 is lower than that of epileptogenic focus (column1).

Then, we systematically show the state of each cortical column versus *K* in Figure 9. Clearly, column1 has been in an epileptic state, and the extreme value differences (peak${}_{E_{1}}$-trough${}_{E_{1}}$ and peak${}_{I_{1}}$-trough${}_{I_{1}}$) of their stable time series are always greater than 0.17 (Figure 9a), and they increase slightly with the increase of *K*. The extreme value differences in column2 and column3 begin to abruptly increase at K=0.2 and K=0.3 (Figure 9a), respectively, suggesting that epileptiform activities propagate to column2 (K≥0.2) and column3 (K≥0.3). In conclusion, the increase of the coupling strength *K* can accelerate and advance the propagation of epileptiform activity and even increase the frequency of the oscillatory state.

### 2.4. Modulation Effects in the Coupled Network Structure

After simulating the epilepsy propagation behavior in the case of typical three-node network motifs, this section focuses on the dynamics of different columns when OS, DBS and electromagnetic induction are applied to the epileptogenic focus (column1).

We firstly consider the modulating effect of OS in the coupled case. Similarly, we set column1 as an epileptic state, while column2 and column3 as a normal low-level state. When *K* is small (K=0.31, Figure 10a,b), a stronger light stimulation (f = 40 Hz, ws = 5 ms, Figure 10b), targeting I${}_{1}$, can effectively suppress the epileptiform activities of all columns compared to Figure 10a (f = 40 Hz, ws = 2 ms). However, column2 and column3 will return back to a more serious epileptic state if we increase *K* to 0.52 (Figure 10c). This requires us to continue to increase the illumination frequency or pulse width to weaken and even eliminate pathological oscillations (f = 100 Hz, ws = 5 ms, Figure 10d; f = 100 Hz, ws = 10 ms, Figure 10e). Notably, when *K* becomes larger, even a constant light stimulation cannot have any effect on the pathosis propagating to column3 (K=0.8, f = 100 Hz, ws = 10 ms, Figure 10f). Furthermore, there is a more systematic analysis of the regulatory effect of OS depicted in Figure 11. The blue area represents the controllable range. If K=0.31 (Figure 11a), a large blue area appears, indicating that three columns can be transferred to the normal background state under appropriate stimulation parameters. With the increase of *K* (K=0.52, Figure 11b; K=0.8, Figure 11c), it is obvious that the blue area greatly decreases until it completely disappears.

Speaking of the dynamical responses of the three columns when DBS targets $I_{1}$, we propose a new indicator, the control rate (CR), which is obtained by calculating the ratio of the number of grids with a peak–trough difference that is less than or equal to 0.05, to the total number of grids. Without loss of generality, we select three coupling strengths (K=0.31, K=0.52, K=0.8) to calculate the CR of each column on the (f, δ) plane and provide a specific statistical histogram (Figure 12). Similar to the results of OS, if *K* is small (K=0.31), the overall control rate of all the columns is relatively high. The larger the value of *K*, the smaller the value of CR, indicating a poorer control effect. The control rate of column3 directly drops to 0 when K=0.8, just like the phenomenon depicted in Figure 11c.

As for the effect of electromagnetic induction, the results are not ideal. Here, the electromagnetic induction was applied to the epileptic focus (column1) at t = 200 ms, and all parameters except $K_{0}$ were consistent with those introduced in Section 4.2. From Figure 13a,d, if $K_{0}\geq 0.1$, electromagnetic induction can always control column1 to a more ideal normal background state. However, electromagnetic induction cannot control column2 and column3. As shown in Figure 13b,e, the values of peak${}_{E_{2}}$-trough${}_{E_{2}}$ and peak${}_{I_{2}}$-trough${}_{I_{2}}$ are always in the yellow band, that is, activities in column2 have not been controlled to the stable equilibrium point state. It should be noted that the blue area in Figure 13b,e (K≤0.2 ) is due to the fact that the epileptic activity does not propagate to column2. Naturally, the electromagnetic induction is also significant for column3 as shown in Figure 13c,f. Again, the blue area (K≤0.3) is due to the fact that the seizure has not yet spread to this column and no further details will be given here.

## 3. Discussions

Based on the Wilson–Cowan framework, this paper compares the mechanisms of OS, DBS and electromagnetic induction on the generation and termination of epilepsy. Under the classical trimodal coupling network structure, the difference between the three strategies in controlling the propagation of epilepsy is further explained.

Firstly, we analyzed the effect of excitatory input (i.e., $P_{E}$ and $P_{I}$) from other brain regions received by the excitatory and inhibitory neuronal populations on their activities. It is found that $P_{E}$ and $P_{I}$ can induce the system to present the bistable phenomenon in which a low-level stable equilibrium state (normal background state) and an oscillatory state (epileptic state) coexist (Figure 1c). This suggests that the transition between these two states can be induced by applying an external stimulus. Moreover, the changes of $P_{E}$ and $P_{I}$ can also induce the system to exhibit Hopf bifurcation and a fold of fixed point bifurcation many times (Figure 1a,b), causing a transition between the interictal and ictal states. Our results support the earlier clinic-based study, that is, the onset and offset of ictal-like discharges are well-defined mathematical events: the Saddle-node and the Hopf bifurcation [41].

Secondly, the regulation mechanisms of OS, DBS and electromagnetic induction for epilepsy were studied and compared. For OS, since studies have shown that inhibitory neurons play an important role in seizures [20,21,22], the inhibitory neuronal population was selected as the stimulation target in our study. However, stimulating only the inhibitory neuronal populations may be impractical for DBS and electromagnetic induction, so their stimulation targets include both excitatory and inhibitory neuronal populations. The results in Section 2.2 show that OS and DBS have similar effects, and both can control the ill-conditioned oscillation mode to a normal low-level state, while electromagnetic induction can only regulate it to a medium-level equilibrium state. Moreover, the neural activities finally regulated by OS and DBS are oscillations with very small amplitudes at the stimulation frequency. Therefore, combining the stimulation effect and the precision of stimulation, OS demonstrates great superiority compared to the other two. After demonstrating that the three-node feedback loop structure can model the propagation behavior of epilepsy well (Section 2.3), we also further verified the advantages of OS based on a typical trimodal coupling network (Section 2.4). Similarly, OS and DBS targeting epileptic focus can not only control the pathological activity of their own regions, but also control the neighboring brain regions of seizure propagation, while electromagnetic induction can only stay in the state of controlling their own brain regions. It is worth noting that DBS and electromagnetic stimulation are established tools for clinical cases and animal experiments. Our analysis can provide a reference for clinicians and experimenters in the selection of stimulus parameters. However, for complex neurological diseases such as epilepsy, OS is still far from clinical application. Our current model results support experimental studies in which OS inhibits seizures in animals. We hope that these results will provide some feasible insights when OS is applied in the clinic [24,25].

The spectral bandwidth of interictal fast epileptic activity was found to be related to the seizure onset zone [42]. Combining power spectrum analysis to refine the conclusions of this paper is worth doing in the future. Transcranial magnetic stimulation has certain positive effects in regulating and measuring the changes of cortical excitability [43]; this paper is only limited to the influence of internal electromagnetic induction. The regulation of external magnetic technology on epilepsy, and its similarities and differences to OS and DBS, need to be further studied. Moreover, the Wilson–Cowan model has also been used to study neurological diseases such as Parkinson’s disease and essential tremor [44,45,46], so it is hoped that these insights will provide guidance for these neuropsychiatric diseases.

## 4. Materials and Methods

### 4.1. Wilson–Cowan Model

With a simple neural circuit framework and equations, the Wilson–Cowan model facilitates the description of the nonlinear dynamics inherent in excitatory–inhibitory interactions in cortical tissue. As shown in Figure 14, excitatory neural populations (E), inhibitory neural populations (I), excitatory synaptic connections (arrows) and inhibitory synaptic connections (lines with filled circles) are considered and the dynamics of the neural population is described as the following differential equations: (7)$$\begin{array}{ccc} \tau_{E}\frac{dE}{dt} & =-E+\left(k_{E}-r_{E}E\right)S_{E}\left(C_{1}E-C_{2}I+P_{E}\right), & \end{array}$$(8)$$\begin{array}{cc} \tau_{I}\frac{dI}{dt} & =-I+\left(k_{I}-r_{I}I\right)S_{I}\left(C_{3}E-C_{4}I+P_{I}\right), \end{array}$$
where E(t) and I(t) are the averaged activities of the excitatory and the inhibitory neural populations, that is, they represent the proportion of excitatory or inhibitory populations firing at time *t*. $\tau_{E}$ and $\tau_{I}$ are synaptic decay associated time constants. $r_{E}$ and $r_{I}$ show the refractory constants. $C_{i}\;\left(i=1,2,3,4\right)$ denote the coupling strengths. $k_{E}$, $k_{I}$ are also constants. These parameters are set as $\tau_{E}=\tau_{I}=8$, $r_{E}=r_{I}=k_{E}=k_{I}=1$, $C_{1}=16$, $C_{2}=12$, $C_{3}=15$, $C_{4}=3$. $P_{E}$ and $P_{I}$ are model external inputs to the excitatory and the inhibitory populations, respectively. Epileptiform activities regulated by $P_{E}$ and $P_{I}$ are studied in Section 2.1. $S_{i}\left(x\right)$ is a sigmoid function:(9)$$\begin{array}{cc} S_{i}\left(x\right) & =\frac{1}{1+\exp(-a_{i}\left(x-\theta_{i}\right))}-\frac{1}{1+\exp(a_{i}\theta )},\;\;\;\;i=E,I. \end{array}$$

The parameters of the sigmoid function are fixed as: $a_{E}=1.3$, $\theta_{E}=4$, $a_{I}=2$, $\theta_{I}=3.7$. All parameter settings are consistent with previous literature [32,47].

### 4.2. Three Neuromodulation Strategies

Three classical neuromodulation strategies are considered in this paper, the first of which is optogenetic stimulation (OS). In past studies, channelrhodopsin-2 (ChR2) has been used as the light-activated channels or transporters to modulate neuronal activity [26,27,48,49]. Considering the fact that the ChR2 current decays bi-exponentially can be explained by the four-state model, here we combine a classical four-state model with the Wilson–Cowan model to study the modulation of epilepsy by optogenetic stimulation. Its dynamic equations are described as:(10)$$\begin{array}{ccc} \frac{dc1}{dt} & =G_{r}c2+G_{d1}o1-k_{1}c1, & \end{array}$$(11)$$\begin{array}{ccc} \frac{do1}{dt} & =k_{1}c1-\left(G_{d1}+e_{12}\right)o1+e_{21}o2, & \end{array}$$(12)$$\begin{array}{ccc} \frac{do2}{dt} & =k_{2}c2-\left(G_{d2}+e_{21}\right)o1+e_{12}o1, & \end{array}$$(13)$$\begin{array}{cc} \frac{dp}{dt} & =\left(S_{0}\left(\theta \right)-p\right)/\tau_{ChR2}, \end{array}$$
where o1+o2+c1+c2=1, o1 and o2 are the probabilities of open state. c1 and c2 are the probabilities of closed state. $G_{d1}$ and $G_{d2}$ are the rates of transition o1→c1 and o2→c2. $G_{r}$ denotes the rate of thermal conversion from c2 to c1. $e_{12}$ and $e_{21}$ show the rates of transition from o1 to o2 and vice versa. $k_{1}$ and $k_{2}$ represent the activation rates of process c1→o1 and c2→o2, respectively. *p* is a state-variable. $S_{0}\left(\theta \right)$ is the time- and irradiance-dependent activation function for ChR2. $\tau_{ChR2}$ is the activation time constant of ChR2. So, the current induced by ChR2 is:(14)$$\begin{array}{cc} I_{ChR2} & =g_{{}_{ChR2}}G\left(V\right)\left(o1+\gamma o2\right)\left(V-E_{{}_{ChR2}}\right), \end{array}$$
and the voltage induced by the ChR2 channels per mini-column is:(15)$$\begin{array}{cc} U_{ChR2} & =I_{ChR2}N_{ChR2}R_{m}. \end{array}$$

Here, $g_{{}_{ChR2}}$ is the max conductance. γ is the ratio of conductances of o2/o1. $E_{ChR2}$ is the reversal potential of ChR2. $N_{ChR2}$ is the number of ChR2 channels per column. Cortical columns—known as hypercolumns, macrocolumns, or modules—are groups of neurons in the cortex that can be successfully penetrated by vertical probes and have nearly identical receptive fields. $R_{m}$ denotes the mean resistance of the targeted population. G(V) is one voltage-dependent rectification function:(16)$$\begin{array}{cc} G(V) & =[1.6408-14.6408\exp(-V/42.7671)]/V. \end{array}$$

It is worth noting that the optogenetic control term is applied to the inhibitory neuronal population, and the relationship between firing activity I and membrane potential V of the inhibitory neuron population is linked by V=αI−70 (α=250) [27]. So, the dynamical Equation (8) is adjusted to Equation (Equation 17) when OS is added:(17)$$\begin{array}{cc} \tau_{I}\frac{dI}{dt} & =-I+\left(k_{I}-r_{I}I\right)S_{I}\left(C_{3}E-C_{4}I+P_{I}-U_{ChR2}/\alpha \right). \end{array}$$

The values of these parameters are consistent with the previous literature [27,48] and are also summarized in Table 1.

The second one is deep brain stimulation (DBS), which is described as the typical period step function in the form of square waves:(18)$$\begin{array}{cc} DBS(t) & =I_{DBS}*heavside\left(\sin\left(2\pi f_{DBS}t\right)\right)*\left(1-heavside\left(\sin\left(2\pi f_{DBS}\left(t+\delta \right)\right)\right)\right), \end{array}$$
where $I_{DBS}$, $f_{DBS}$ and δ are the amplitude, frequency and positive duration of pulse, respectively. Due to the small number of inhibitory neurons in the cerebral cortex, it is difficult for the nucleus-targeted DBS technology to control only the inhibitory neuron population. Hence, the DBS in this paper is applied to neural populations *E* and *I* at the same time, and the corresponding dynamical Equations (Equation 7) and (8) are adjusted to Equations (Equation 19) and (20), respectively: (19)$$\begin{array}{ccc} \tau_{E}\frac{dE}{dt} & =-E+\left(k_{E}-r_{E}E\right)S_{E}\left(C_{1}E-C_{2}I+P_{E}-DBS\left(t\right)\right), & \end{array}$$(20)$$\begin{array}{cc} \tau_{I}\frac{dI}{dt} & =-I+\left(k_{I}-r_{I}I\right)S_{I}\left(C_{3}E-C_{4}I+P_{I}-DBS\left(t\right)\right). \end{array}$$

As we know, the electromagnetic effects can vary the intercellular and extracellular ion concentrations and further influence the membrane voltage. So the third case focuses on the dynamics of epileptiform behaviors under the influence of electromagnetic induction. The dynamical model proposed by Ma and his colleagues was chosen for our study [50]. The mathematical description is as follows:(21)$$\begin{array}{ccc} I_{mfi} & =K_{0}\rho \left(\phi_{i}\right)V_{i}, & \end{array}$$(22)$$\begin{array}{ccc} \frac{d\phi_{i}}{dt} & =K_{1}V_{i}-K_{2}\phi_{i}, & \end{array}$$(23)$$\begin{array}{cc} \rho (\phi_{i}) & =\alpha_{1}+3\beta_{1}\phi_{i}^{2}. \end{array}$$

Here, i=E or *I*. $\phi_{i}$ describes the average magnetic flux across membrane. $K_{0}$ is the feedback gain from magnetic flux, which directly determines the effect of electromagnetic induction on neural populations. $K_{1}$ is the gain from the target population. $K_{2}$ is the gain of self-inductance effect. $\alpha_{1}$ and $\beta_{1}$ are constants. In the following sections, we systematically analyze a series of dynamical phenomena induced by $k_{0}$. Other parameters are fixed as: $K_{1}=0.01$, $K_{2}=0.1$, $\alpha_{1}=0.1$, $\beta_{1}=0.07$. In addition, electromagnetic effects are considered to apply to E and I simultaneously. $V_{E}=\alpha E-70$ and $V_{I}=\alpha I-70$ (α=250) [27] are used to scale the inhibitory neural activities to membrane potentials. Therefore, when electromagnetic induction is introduced to the system, Equations (Equation 7) and (8) are changed to Equations (Equation 24) and (25), respectively: (24)$$\begin{array}{ccc} \tau_{E}\frac{dE}{dt} & =-E+\left(k_{E}-r_{E}E\right)S_{E}\left(C_{1}E-C_{2}I+P_{E}-I_{mfE}\right), & \end{array}$$(25)$$\begin{array}{cc} \tau_{I}\frac{dI}{dt} & =-I+\left(k_{I}-r_{I}I\right)S_{I}\left(C_{3}E-C_{4}I+P_{I}-I_{mfI}\right). \end{array}$$

All the numerical simulations in this paper were carried out in the MATLAB (MathWorks, Natick, MA, USA) simulating environment. The algorithm used was the classic Runge–Kutta method with adaptive steps, and was verified with the help of XPPAUT software [51]. Diagrams for dynamical bifurcation mechanisms were created under XPPAUT-AUTO. The initial values were: E(0)=0.11, I(0)=0.09, o1(0)=0.4, o2(0)=0.4, c1(0)=0.1, p(0)=0, $\phi_{E}\left(0\right)=0.5$ and $\phi_{i}\left(0\right)=0.5$.

## Figures and Tables

**Figure 1 ijms-23-13652-f001:**
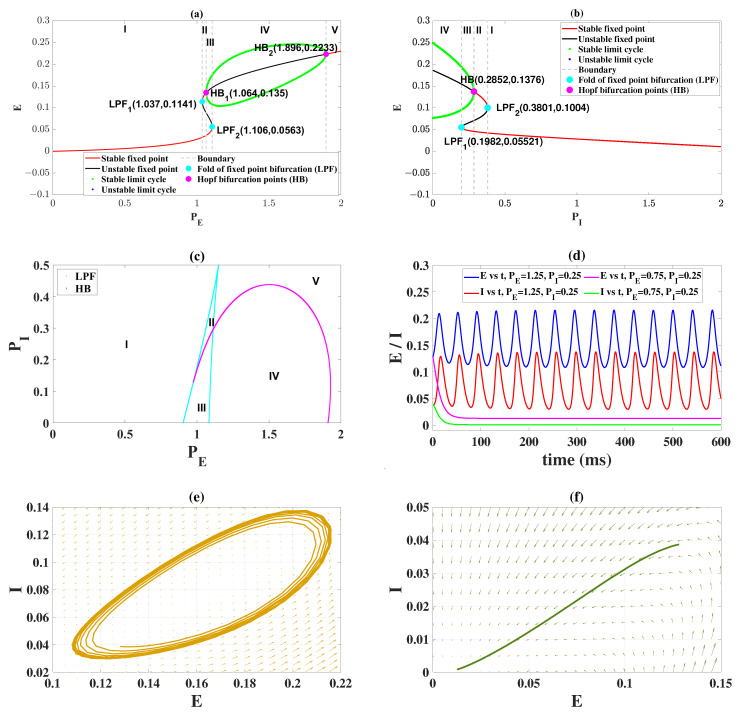
Dynamical bifurcation diagrams induced by $P_{E}$ (**a**) and $P_{I}$ (**b**). Two-dimensional bifurcation diagrams on the plane ($P_{E}$, $P_{I}$) (**c**). Time series of normal background state and oscillating state by varying $P_{E}$ and $P_{I}$ (**d**). The phase diagram corresponding to the two states: oscillating state (**e**) and normal background state (**f**). The meanings of the symbols are as follows: the low-level stable equilibrium state exists (I), the low-level stable equilibrium state and the medium-level stable equilibrium state coexist (II), the stable limit cycle and the low-level stable equilibrium state coexist (III), the stable limit cycle (oscillating state) exists (IV), the high-level stable equilibrium state exists (V).

**Figure 2 ijms-23-13652-f002:**
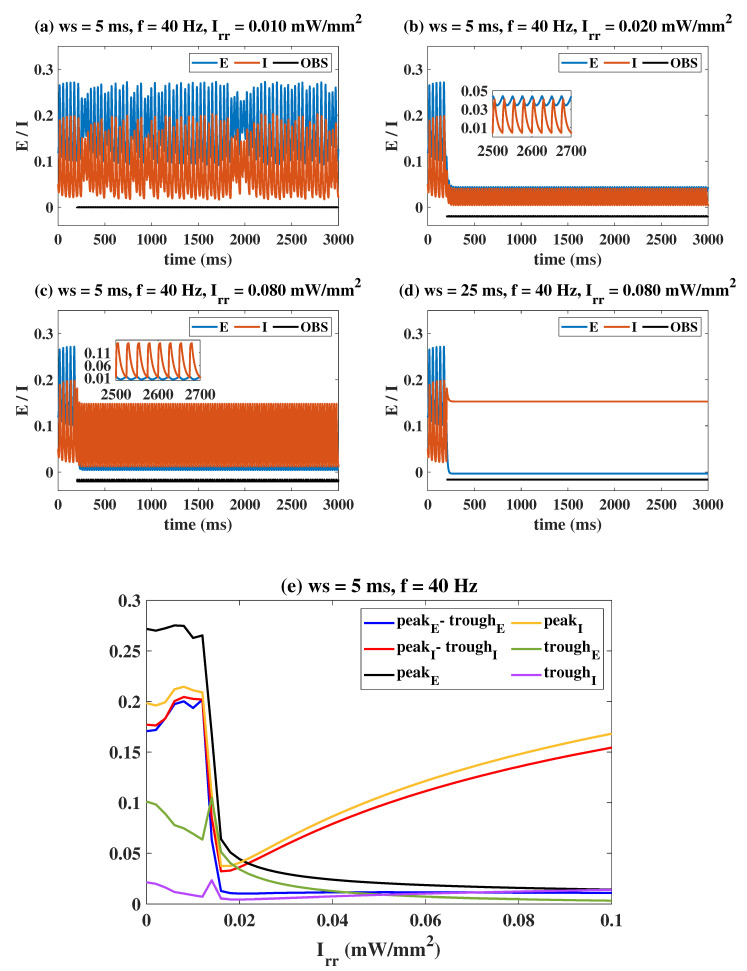
(**a**–**d**) Typical time series under the influence of OS. (**e**) The evolution of the observed quantitative indicators with I${}_{rr}$. Figures (**a**,**b**) represent the OS being invalid and valid, respectively. Figures (**c**,**d**) show that the effect of OS on E is desired, but the effect on I is not. The partial, enlarged views in Figures (**a**–**d**) are also given for clarity. Peak${}_{E}$ and trough${}_{E}$ are the maxima and minima of the stabilized time series of population E, respectively. The definitions of Peak${}_{I}$ and trough${}_{I}$ for population I are similar.

**Figure 3 ijms-23-13652-f003:**
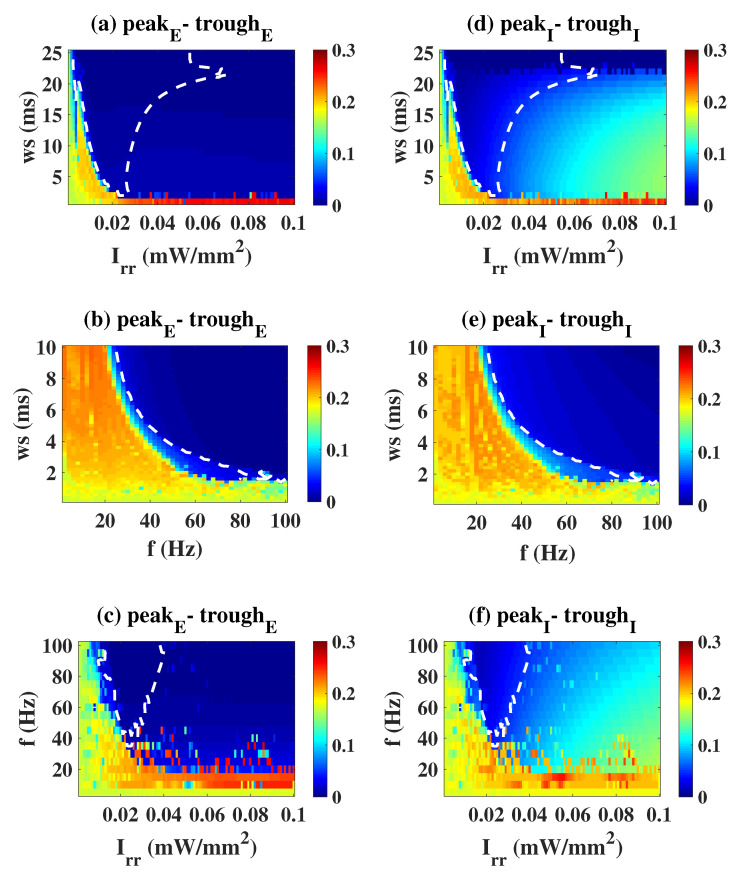
Evolutions of the observed indicators on the two-stimulus parameter plane. Areas enclosed by the white dashed line are the effective parameter area (peak, trough, peak-trough of E and I are all smaller than 0.05). (**a**,**d**): f = 40 Hz, (**b**,**e**): I${}_{rr}$ = 0.015 mW/mm${}^{2}$, (**c**,**f**): ws = 2 ms.

**Figure 4 ijms-23-13652-f004:**
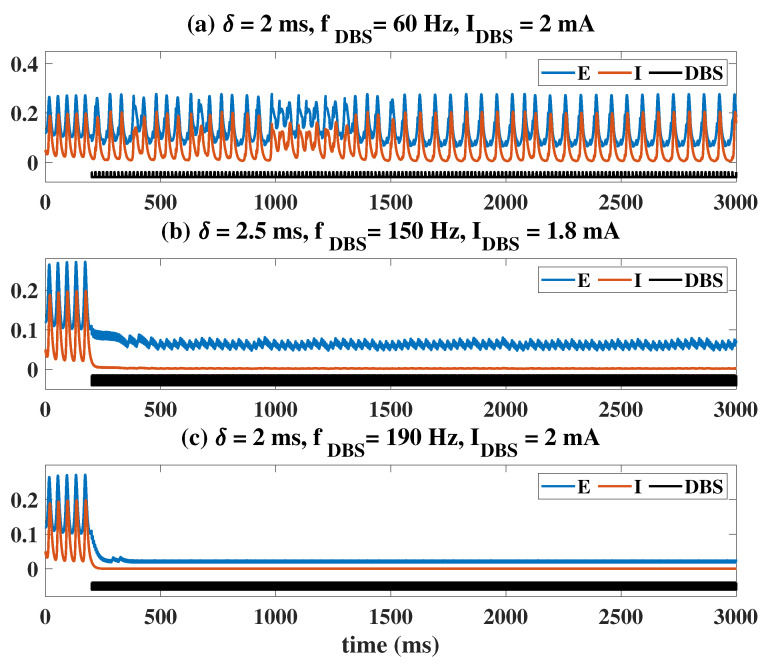
Control effects of DBS under different parameter settings. The control effect shown in (**c**) is better than that of (**a**,**b**).

**Figure 5 ijms-23-13652-f005:**
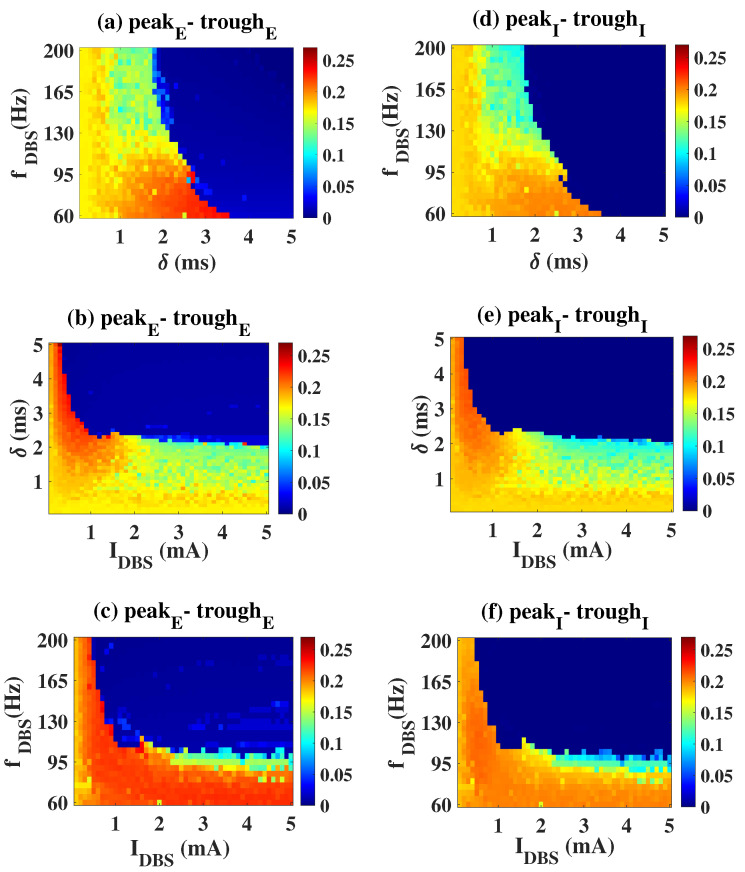
Evolutions of the observed indicators on the two-stimulus parameter plane. The dark blue area represents the stimulus effective area. (**a**,**d**): I${}_{DBS}$ = 2 mA, (**b**,**e**): f${}_{DBS}$ = 120 Hz, (**c**,**f**): δ = 2.5 ms.

**Figure 6 ijms-23-13652-f006:**
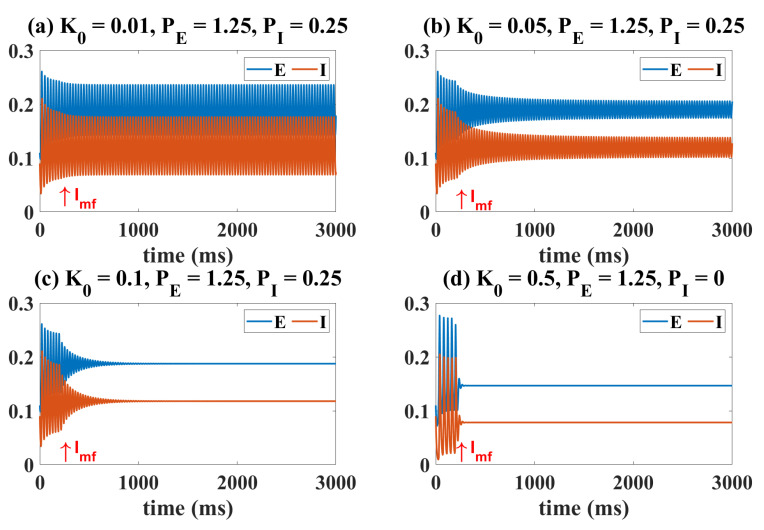
Time series diagram under the influence of electromagnetic induction. (**a**) $I_{mf}$ is ineffective, i.e., the amplitude of the oscillatory state hardly changes. As $K_{0}$ increases, amplitude of the oscillatory state decreases (**b**) and eventually becomes the high-level (**c**) and the medium-level equilibrium point states (**d**). $I_{mf}$ and arrows indicate that the electromagnetic induction model and Wilson–Cowan model are coupled at t = 200 ms.

**Figure 7 ijms-23-13652-f007:**
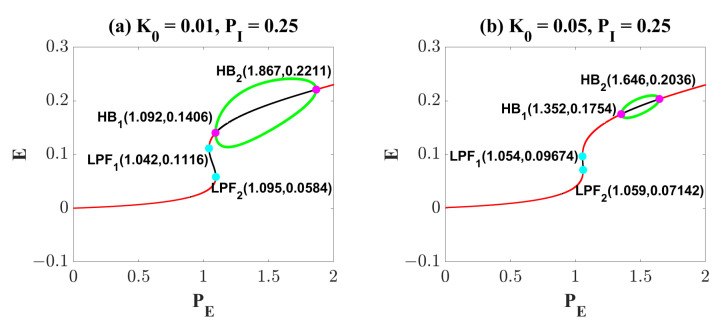
Dynamical bifurcation diagrams induced by $P_{E}$ under the influence of electromagnetic induction (**a**–**f**). Two-dimensional bifurcation diagrams on the plane ($P_{E}$, $P_{I}$) under the influence of electromagnetic induction (**g**–**i**). All symbols have the same meaning as in Figure 1. An increase in $K_{0}$ reduces the region of the oscillatory state (IV).

**Figure 8 ijms-23-13652-f008:**
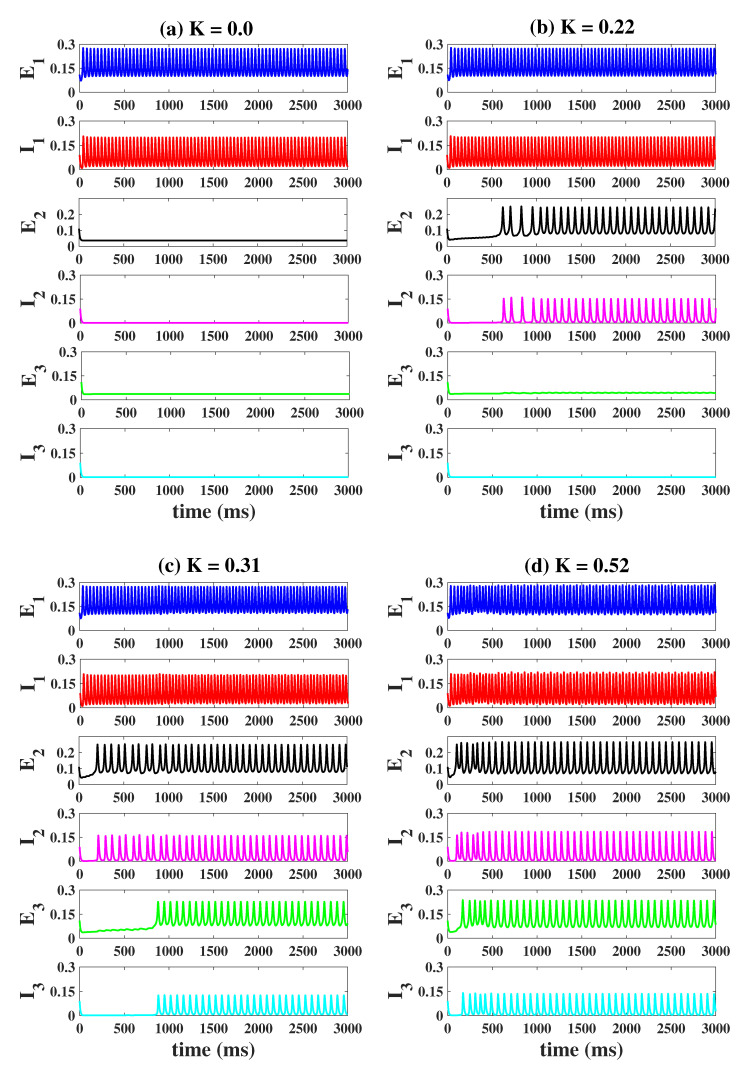
Time series of the three columns under different coupling connection strengths *K*. An increase in *K* will cause columns 2 and 3 to oscillate gradually (**b**–**d**), and column2 will oscillate before column3 (**b**).

**Figure 9 ijms-23-13652-f009:**
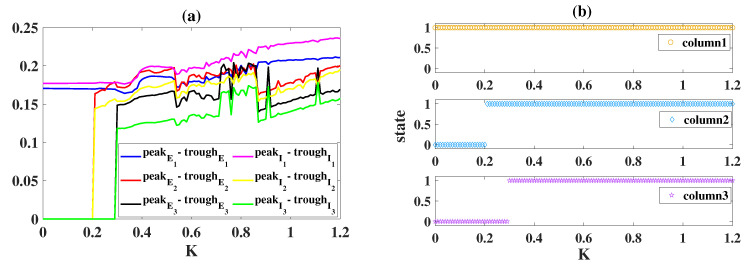
(**a**) Relationship of the extreme value difference with *K*. (**b**) Relationship of the system’s state with *K*. When K≥0.2, column2 starts to oscillate, that is, from non-oscillating state 0 to oscillating state 1. Column3 starts to oscillate from K=0.3.

**Figure 10 ijms-23-13652-f010:**
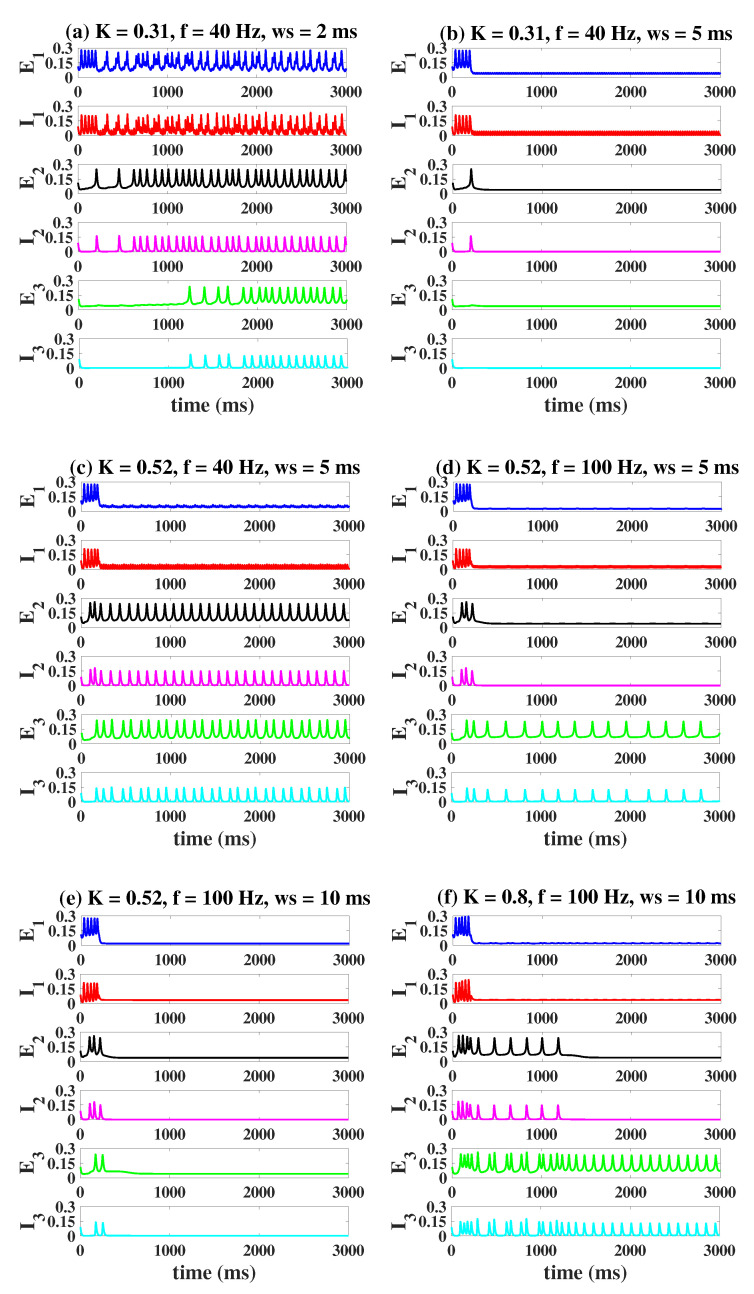
Under the different coupling strengths *K*, time series of the three columns when OS targets $I_{1}$ with different stimulus parameters. When *K* is small (K=0.31, K=0.52), the proper OS can always control all the three columns (**b**,**e**). However, if the value of *K* is larger (K=0.8), column3 cannot always be controlled regardless of the settings of the stimulus parameters (**f**).

**Figure 11 ijms-23-13652-f011:**
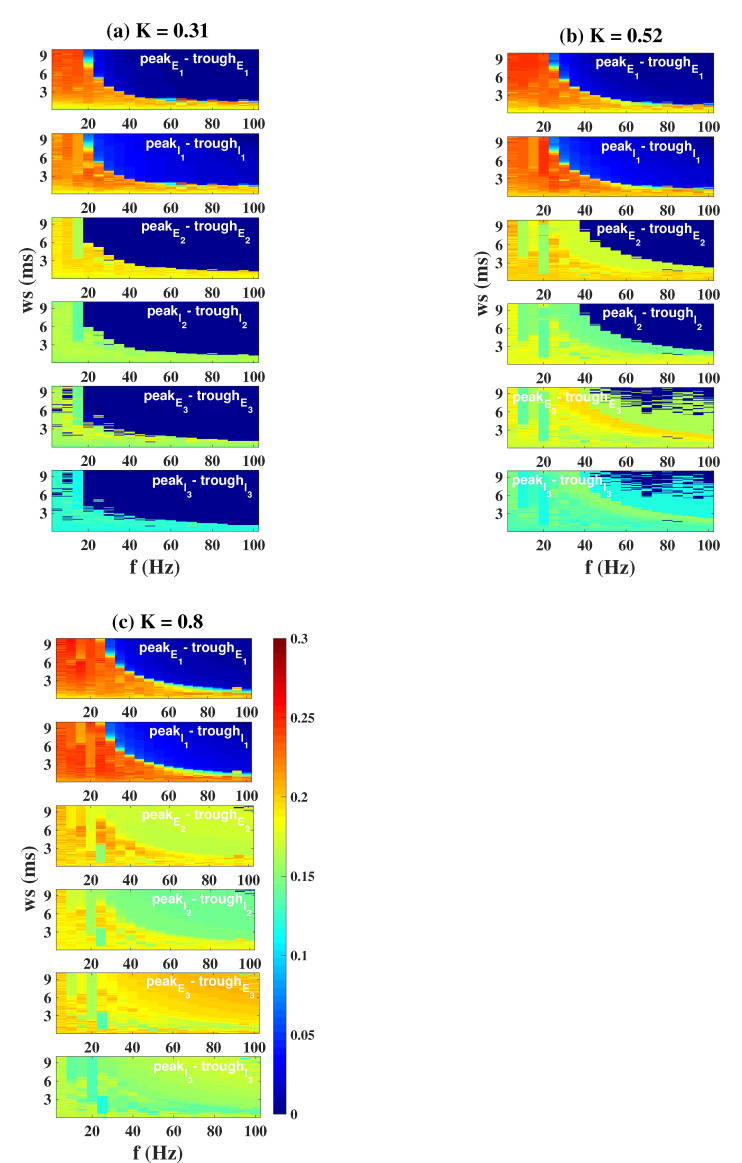
Dynamical responses of the three columns when OS targets $I_{1}$. The control effect of column2 and column3 is closely related to K. Three compartments can be controlled well when K=0.31 (**a**). The controllable areas of column2 and column3 are reduced (K=0.52, (**b**)), even to zero (K=0.8, (**c**)).

**Figure 12 ijms-23-13652-f012:**
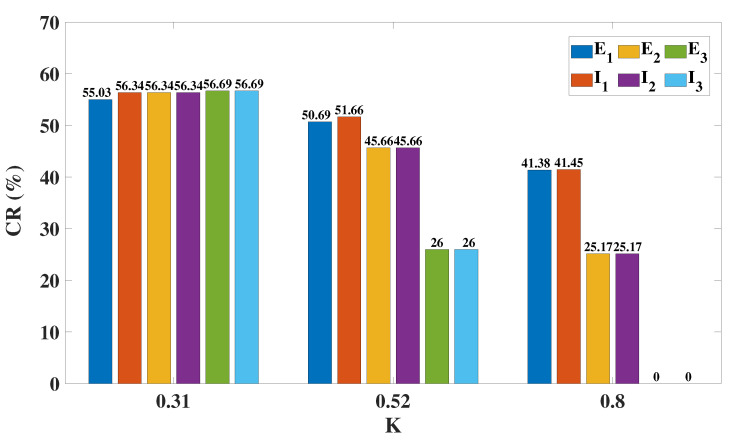
Control rate of the three columns, CR(%), when DBS targets column1. The higher the CR(%) value, the better the control effect. Regardless of the value of *K*, column1 is well controlled. When K=0.52, the CR of column2 is slightly reduced, and the CR of column3 is reduced by half. When K=0.8, column2 also has a significant reduction in CR, and column3 has a zero CR.

**Figure 13 ijms-23-13652-f013:**
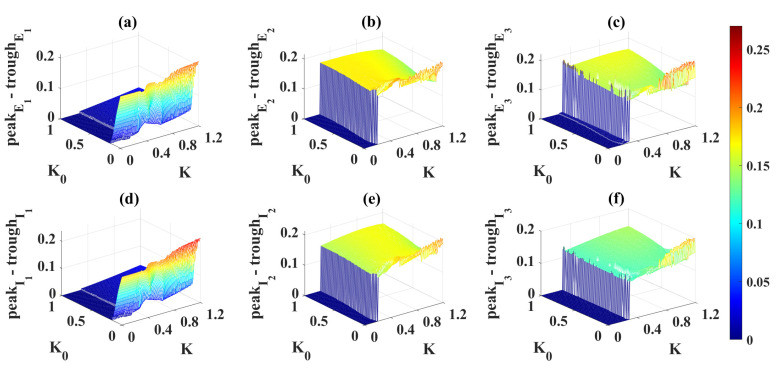
Dynamical responses of the three columns ((**a**,**d**): column1, (**b**,**e**): column2, (**c**,**f**): column3) under the influence of different coupling strengths *K* and electromagnetic induction feedback gains $K_{0}$. The electromagnetic induction applied in column1 appears to be unable to control column2 (**b**,**e**) and column3 (**c**,**f**), which are in an epileptic state due to the propagation effects.

**Figure 14 ijms-23-13652-f014:**
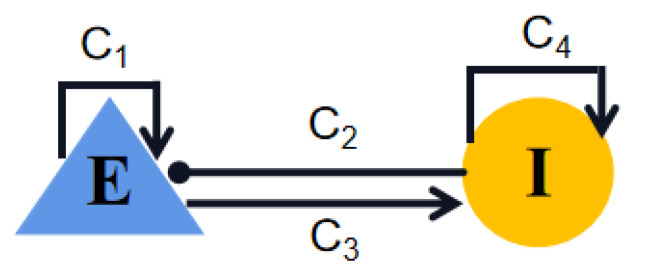
Schematic diagram of the Wilson–Cowan model. E and I are the excitatory and inhibitory neural populations, respectively. $C_{i}\;\left(i=1,2,3,4\right)$ represent the coupling strengths between and within the subpopulations. Excitatory projections are denoted by arrows, while lines with filled circle represent inhibitory projections.

**Table 1 ijms-23-13652-t001:** Brief descriptions and baseline values of optogenetic stimulation model parameters.

Symbol	Interpretation	Value
$k_{1}$	activation rates of process c1→o1	$\varepsilon_{1}*F*p$
$k_{2}$	activation rates of process c2→o2	$\varepsilon_{2}*F*p$
$G_{d1}$	transition rate of o1→c1	0.075+0.043∗tanh(−(V+20)/20)
$G_{d2}$	transition rate of o2→c2	0.05
$e_{12}$	transition rate of o1→o2	$0.011+0.005*\ln(1+I_{rr}/0.024)$
$e_{21}$	transition rate of o2→o2	$0.008+0.004*\ln(1+I_{rr}/0.024)$
$G_{r}$	thermal recovery rate of c2→c1	$4.34587*10^{5}*exp\left(-0.0211539274*V\right)$
$\varepsilon_{1}$	quantum efficiency of light absorption in c1 state	0.8535
$\varepsilon_{2}$	quantum efficiency of light absorption in c2 state	0.14
*F*	photon flux at a given time	$F=\sigma_{ret}*I_{rr}*\lambda /\left(w_{loss}*hc\right)$
$\sigma_{ret}$	effective cross section of a ChR2 channel	$10^{-20}$ m${}^{2}$
$I_{rr}$	light irradiance	0–10 mW/mm${}^{2}$
λ	light wavelength	470 nm
$w_{loss}$	scaling factor of photon loss	1.3
*h*	Planck’s constant	$6.626\times 10^{-34}$ Js
*c*	light speed	$3\times 10^{8}$ m/s
$\tau_{ChR2}$	activation time constant of ChR2	1.3 ms
$S_{0}\left(\theta \right)$	activation function of ChR2	$0.5*(1+tanh\left(120\left(100I_{rr}-0.1\right)\right))$
$g_{{}_{ChR2}}$	max conductance of ChR2	0.4 mS/cm${}^{2}$
γ	ratio of conductances of o2/o1	0.1
$E_{ChR2}$	reversal potential of ChR2	0

## Data Availability

The data supporting the reported results has been included in the paper.

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
