# Peer review of "Dynamical Mechanism Analysis of Three Neuroregulatory Strategies on the Modulation of Seizures"

_ijms, 2022, doi:10.3390/ijms232113652_

Round 1

Reviewer 1 Report

ijms-1918386: Dynamical mechanism analysis of three neuroregulatory strategies on the modulation of seizures

In this clearly written manuscript, the authors develop a new model of epilepsy regulation by use of three different approaches: optogenetic stimulation (OS), deep brain stimulation (DBS) and electromagnetic induction. All essential mathematical modifications are described successively, cogently and conclusions are supported with obtained data. This manuscript could be recommended for publication after a few corrections.

Remarks/recommendations:

1)     “The current common treatments for it include…” (page 1);

2)      “Among which, neural stimulation with the aid of electric or magnetic technologies has been the research focus in the field of neuroscience over the past century” should be rewritten for clarity (page 1);

3)     In Fig. 1 legend, “lines with filled circles” should be in singular form (page 3);

4)     No explanation what “columns” mean (page 4 and below);

5)     “The second one is the deep brain stimulation (DBS),…” (page 4);

6)     “Here, i = E or I“ would be more suitable (page 5);

7)     Fig. 3 legend needs more detailed description (page 8);

8)     In “And the partial enlarged views are also given for clarity”, “And” should be omitted (page 8);

9)     The legends to figures 5 and 13 need detailed description (pages 10 and 17, respectively);

10)  “not insignificant” might be replace by “significant” (page 18);

11)  English needs to be double checked.

Author Response

In this clearly written manuscript, the authors develop a new model of epilepsy regulation by use of three different approaches: optogenetic stimulation (OS), deep brain stimulation (DBS) and electromagnetic induction. All essential mathematical modifications are described successively, cogently and conclusions are supported with obtained data. This manuscript could be recommended for publication after a few corrections.

Point 1: ”The current common treatments for it include…” (page 1).

Response 1: Thank you so much for your careful check. The word “including” in this sentence has been replaced with the word “include”.

Point 2: “Among which, neural stimulation with the aid of electric or magnetic technologies has been the research focus in the field of neuroscience over the past century” should be rewritten for clarity (page 1).

Response 2: We sincerely thank you for your valuable suggestion. We have changed this sentence to “In terms of neuroregulatory strategies, electrical and magnetic technologies have been the research focus over the past century ”.

Point 3:  In Fig. 1 legend, “lines with filled circles” should be in singular form (page 3).

Response 3: Thank you so much for your careful check. The word “circles” in this sentence has been replaced with the word “circle”.

Point 4: No explanation what “columns” mean (page 4 and below)

Response 4: Thank you for your efforts. We add an explanation of the cortical column on page 4:

“Cortical columns, known as hypercolumns, macrocolumns, or modules, are groups of neurons in the cortex that can be successfully penetrated by vertical probes and have nearly identical receptive fields”.

Point 5:  “The second one is the deep brain stimulation (DBS),…” (page 4).

Response 5: Thank you so much for your careful check. We revise it in the newly submitted manuscript.

Point 6:  “Here, i = E or I“ would be more suitable (page 5).

Response 6: Thank you so much for your valuable suggestion. We revise it in the newly submitted manuscript.

Point 7: Fig. 3 legend needs more detailed description (page 8).

Response 7: Thank you so much for your valuable comments. We rich it in the newly submitted manuscript as follows:

Fig.3. (a-d) Typical time series under the influence of OS. (e) The evolution of the observed quantitative indicators with I$ _{rr} $. Figures (a) and (b) represent the OS is invalid and valid, respectively. Figures (c) and (d) show that the effect of OS on E is desired, but the effect on I is not. The partial enlarged views in Figures (a) to (d) are also given for clarity. Peak$ _E $ and trough$_E $ are the maxima and minima of the stabilized time series of population E, respectively. The definitions of Peak$ _I $ and trough$_I $ for population I are similar.

Point 8: In “And the partial enlarged views are also given for clarity”, “And” should be omitted (page 8).

Response 8: Thank you so much for your valuable suggestion. We omit it in the newly submitted manuscript.

Point 9: The legends to figures 5 and 13 need detailed description (pages 10 and 17, respectively).

Response 9: Thank you so much for your valuable suggestion. We rich them in the newly submitted manuscript as follows:

Fig.5. Control effects of DBS under different parameter settings. (a) $ \delta$ = 2 ms, f$_{DBS}$ = 60 Hz, I$_{DBS}$ = 2 mA. (b) $ \delta$ = 2.5 ms, f$_{DBS}$ = 150 Hz, I$_{DBS}$ = 1.8 mA. (c) $ \delta$ = 2 ms, f$_{DBS}$ = 190 Hz, I$_{DBS}$ = 2 mA. The control effect shown in figure (c) is better than that of (a) and (b).

Fig.13. Control rate of the three columns, CR(\%), when DBS targets the column1. The higher the CR(\%) value, the better the control effect. Regardless of the value of $ K $, the column1 is well controlled. When $ K=0.52 $, CR of the column2 is slightly reduced, and CR of the column3 is reduced by half. When $ K=0.8 $, column2 also has a significant reduction in CR, and column3 has a zero CR.

Point 10: “not insignificant” might be replace by “significant” (page 18).

Response 10: Thank you so much for your valuable suggestion. We replace it in the newly submitted manuscript.

Point 11:  English needs to be double checked.

Response 11: We sincerely thank you for your valuable comments and suggestions, and have made a comprehensive revision. Hope the improved manuscript will meet with approval.

Reviewer 2 Report

The manuscript is interesting, it offers many results that must be discussed independently, in order that they can be properly interpreted, not just discussed them very briefly in the Conclusion chapter.

Author Response

The manuscript is interesting, it offers many results that must be discussed independently, in order that they can be properly interpreted, not just discussed them very briefly in the Conclusion chapter.

Response : We sincerely thank you for your valuable comments and suggestions. We enrich the description of the Conclusions and Discussions chapter.

The following description is added to the second paragraph: Our results support the earlier clinic-based study, that is, the onset and offset of ictal-like discharges are well-defined mathematical events: the Saddle-node and the Hopf bifurcation [47].

The following description is added to the third paragraph: It is worth noting that DBS and electromagnetic stimulation are established tools for clinical cases and animal experiments. Our analysis can provide a reference for clinicians and experimenters in the selection of stimulus parameters. However, for complex neurological diseases such as epilepsy, OS is still far from clinical application. Our current model results support experimental studies in which OS inhibits seizures in animals. We hope that these results will provide some feasible insights when OS is applied to the clinic[24, 25].

The last paragraph is rewritten as: Spectral bandwidth of interictal fast epileptic activity was found to be related to the seizure onset zone[48]. Combining power spectrum analysis to refine the conclusions of this paper is worth doing in the future. Transcranial magnetic stimulation has certain positive effects in regulating and measuring the changes of cortical excitability[49], this paper is only limited to the influence of internal electromagnetic induction. The regulation of external magnetic technology on epilepsy, and its similarities and differences with OS and DBS need to be further studied. Moreover, the Wilson-Coman model has also been used to study neurological diseases such as Parkinson’s disease and essential tremor[50, 51, 52]. So it is hoped that these insights will provide guidance for these neuropsychiatric diseases.

Reviewer 3 Report

The authors model the effect of optic stimulation, deep brain stimulation and electromagnetic stimulation on epileptic activity based on the effect on recurrent excitatory and inhibitory neural networks using a Wilson-Cowan model. From a clinical perspective it is highly interesting to better understand the effects of these different stimulation techniques on neuronal populations considering two different populations. It is a well structured manuscript, but writing needs some improvements.

Detailed comments:

Abstract: “infected brain area” I believe that the authors mean the area of seizure propagation.

Discussion/Conclusion

The discussion remains short and superficial and is mainly a repetition of results. I suggest that the authors discuss their findings in the context of prior clinical research e.g. Hopf-bifurcations as a model for the transition between interictal and ictal states [Jirsa et al., 2014] and spectral bandwidth changes of interictal fast epileptic activity as a model of changes in dampening of epileptic activity [Heers et al., 2018].

The authors should discuss differences between animal and clinical research as there are e.g. it is impossible to apply OS in clinical practice, whereas DBS and electromagnetic stimulation are established clinical tools.

p. 19, paragraph 1:“… and then cause the generation and termination of the epilepsy.”

There seems to be a misconception of epilepsy: I believe that the authors want to describe the transition between interictal and ictal state.

“…, but also control the diseased brain regions they infect, …”

I believe that the authors mean spreading of seizure activity to neighbouring brain regions (see also Abstract above)

The writing of the manuscript needs to be carefully reviewed for redundancies to improve readability. Figure legends could comprise text which guides the authors in the interpretation of the presented findings.

References

Heers M, Helias M, Hedrich T, Dümpelmann M, Schulze-Bonhage A, Ball T (2018): Spectral bandwidth of interictal fast epileptic activity characterizes the seizure onset zone. NeuroImage Clin 17:865–872.

Jirsa VK, Stacey WC, Quilichini PP, Ivanov AI, Bernard C (2014): On the nature of seizure dynamics. Brain 137:2210–2230.

Author Response

The authors model the effect of optic stimulation, deep brain stimulation and electromagnetic stimulation on epileptic activity based on the effect on recurrent excitatory and inhibitory neural networks using a Wilson-Cowan model. From a clinical perspective it is highly interesting to better understand the effects of these different stimulation techniques on neuronal populations considering two different populations. It is a well structured manuscript, but writing needs some improvements.

Point 1: Abstract: ”infected brain area” I believe that the authors mean the area of seizure propagation.

Response 1: Thank you so much for your valuable comments. We revise it in the newly submitted manuscript: ..., OS and DBS applied to the epileptic focus play similar positive roles in controlling the behavior of the area of seizure propagation, ...

Point 2: Discussion/Conclusion: The discussion remains short and superficial and is mainly a repetition of results. I suggest that the authors discuss their findings in the context of prior clinical research e.g. Hopf-bifurcations as a model for the transition between interictal and ictal states [Jirsa et al., 2014] and spectral bandwidth changes of interictal fast epileptic activity as a model of changes in dampening of epileptic activity [Heers et al., 2018].

The authors should discuss differences between animal and clinical research as there are e.g. it is impossible to apply OS in clinical practice, whereas DBS and electromagnetic stimulation are established clinical tools.

References

Heers M, Helias M, Hedrich T, Dümpelmann M, Schulze-Bonhage A, Ball T (2018): Spectral bandwidth of interictal fast epileptic activity characterizes the seizure onset zone. NeuroImage Clin 17:865–872.

Jirsa VK, Stacey WC, Quilichini PP, Ivanov AI, Bernard C (2014): On the nature of seizure dynamics. Brain 137:2210–2230.

Response 2: We sincerely thank you for your valuable suggestion. We enrich the description of the Conclusions and Discussions chapter.

The following description is added to the second paragraph: Our results support the earlier clinic-based study, that is, the onset and offset of ictal-like discharges are well-defined mathematical events: the Saddle-node and the Hopf bifurcation [47].

The following description is added to the third paragraph: It is worth noting that DBS and electromagnetic stimulation are established tools for clinical cases and animal experiments. Our analysis can provide a reference for clinicians and experimenters in the selection of stimulus parameters. However, for complex neurological diseases such as epilepsy, OS is still far from clinical application. Our current model results support experimental studies in which OS inhibits seizures in animals. We hope that these results will provide some feasible insights when OS is applied to the clinic[24, 25].

The last paragraph is rewritten as: Spectral bandwidth of interictal fast epileptic activity was found to be related to the seizure onset zone[48]. Combining power spectrum analysis to refine the conclusions of this paper is worth doing in the future. Transcranial magnetic stimulation has certain positive effects in regulating and measuring the changes of cortical excitability[49], this paper is only limited to the influence of internal electromagnetic induction. The regulation of external magnetic technology on epilepsy, and its similarities and differences with OS and DBS need to be further studied. Moreover, the Wilson-Coman model has also been used to study neurological diseases such as Parkinson’s disease and essential tremor[50, 51, 52]. So it is hoped that these insights will provide guidance for these neuropsychiatric diseases.

Point 3:  p. 19, paragraph 1:“… and then cause the generation and termination of the epilepsy.” There seems to be a misconception of epilepsy: I believe that the authors want to describe the transition between interictal and ictal state.

Response 3: Thank you so much for your valuable suggestion. We revise it in the newly submitted manuscript: ..., causing a transition between the interictal and ictal state.

Point 4: “…, but also control the diseased brain regions they infect, …”

I believe that the authors mean spreading of seizure activity to neighbouring brain regions (see also Abstract above)

Response 4: Thank you for your efforts. We revise it in the newly submitted manuscript: ..., but also control the neighbouring brain regions of seizure propagation,...

Point 5: The writing of the manuscript needs to be carefully reviewed for redundancies to improve readability. Figure legends could comprise text which guides the authors in the interpretation of the presented findings.

Response 5: We sincerely thank you for your valuable comments and suggestions, and have made a comprehensive revision. Hope the improved manuscript will meet with approval.

Round 2

Reviewer 3 Report

I believe the presented study is an interesting contribution to the field.  The points raises during the review were addressed properly by the authors. So I can fully support its publication in the Intern. J of Molecular Sciences.